

# Uncertainty in projections of future lake thermal dynamics is differentially driven by lake and global climate models

Jacob H. Wynne[1,2], Whitney Woelmer[1], Tadhg N. Moore[1], R. Quinn Thomas[1,3], Kathleen C. Weathers[4] and Cayelan C. Carey[1]

[1] Department of Biological Sciences, Virginia Polytechnic Institute and State University (Virginia Tech), Blacksburg, VA, United States of America
[2] Department of Microbiology, Oregon State University, Corvallis, OR, United States of America
[3] Department of Forest Resources and Environmental Conservation, Virginia Polytechnic Institute and State University (Virginia Tech), Blacksburg, VA, United States of America
[4] Cary Institute of Ecosystem Studies, Millbrook, NY, United States of America

Corresponding author
Whitney Woelmer, wwoelmer@vt.edu

## ABSTRACT

Freshwater ecosystems provide vital services, yet are facing increasing risks from global change. In particular, lake thermal dynamics have been altered around the world as a result of climate change, necessitating a predictive understanding of how climate will continue to alter lakes in the future as well as the associated uncertainty in these predictions. Numerous sources of uncertainty affect projections of future lake conditions but few are quantified, limiting the use of lake modeling projections as management tools. To quantify and evaluate the effects of two potentially important sources of uncertainty, lake model selection uncertainty and climate model selection uncertainty, we developed ensemble projections of lake thermal dynamics for a dimictic lake in New Hampshire, USA (Lake Sunapee). Our ensemble projections used four different climate models as inputs to five vertical one-dimensional (1-D) hydrodynamic lake models under three different climate change scenarios to simulate thermal metrics from 2006 to 2099. We found that almost all the lake thermal metrics modeled (surface water temperature, bottom water temperature, Schmidt stability, stratification duration, and ice cover, but not thermocline depth) are projected to change over the next century. Importantly, we found that the dominant source of uncertainty varied among the thermal metrics, as thermal metrics associated with the surface waters (surface water temperature, total ice duration) were driven primarily by climate model selection uncertainty, while metrics associated with deeper depths (bottom water temperature, stratification duration) were dominated by lake model selection uncertainty. Consequently, our results indicate that researchers generating projections of lake bottom water metrics should prioritize including multiple lake models for best capturing projection uncertainty, while those focusing on lake surface metrics should prioritize including multiple climate models. Overall, our ensemble modeling study reveals important information on how climate change will affect lake thermal properties, and also provides some of the first analyses on how climate model selection uncertainty and lake model selection uncertainty interact to affect projections of future lake dynamics.

**Subjects** Computational Biology, Ecosystem Science, Climate Change Biology,
Freshwater Biology, Environmental Impacts
**Keywords** Ecosystem modeling, Uncertainty, Lake thermal dynamics, Climate change, Scenarios,
Hydrodynamics, Process-based models, Lake models, Ensemble modeling, Management

## INTRODUCTION

Freshwater ecosystems provide essential ecosystem services, including water for drinking, irrigation, and fisheries, and substantial cultural and economic value (*Janssen et al., 2021*). However, freshwater ecosystems have been severely affected by human activities (*IPCC, 2021*), with abrupt and severe water quality degradation occurring in response to climate change (*Woolway et al., 2019*; *Ho & Michalak, 2020*), which is predicted to accelerate in the future (*Weyhenmeyer, Westöö & Willén, 2008*; *Sharma et al., 2019*; *Woolway & Merchant, 2019*). Thus, there is an increasing need for model projections that represent future lake ecosystem conditions to help decision-makers anticipate, prepare for, and potentially mitigate changes in lake ecosystem services (*Brookes et al., 2014*; *Khan et al., 2015*).

Among lake water quality metrics, lake thermal structure (which encompasses water column temperatures, stratification, and duration of ice coverage) plays a key role in lake ecosystem functioning and is extremely sensitive to altered climate (*O'Reilly et al., 2015*; *Woolway & Merchant, 2018*; *Sharma et al., 2021b*; *Woolway, Sharma & Smol, 2022*). For example, thermal stratification (*i.e.,* the presence of a strong temperature gradient from the surface to the bottom of the lake) directly influences mixing regimes (the yearly pattern of thermal stratification; *Lewis Jr, 1983*; *Wetzel, 2001*), which are expected to shift in lakes under most climate change scenarios (*Woolway & Merchant, 2019*). For example, in many temperate, dimictic lakes, *Woolway & Merchant (2019)* projected a shift from two mixing events annually to a single mixing event as lakes lose ice cover. Mixing regimes have major implications for lake ecological processes such as primary productivity, availability of fish habitat, nutrient availability, and atmospheric gas exchange (*Wetzel, 2001*; *Kirillin, 2010*; *Richardson et al., 2017*). In addition, the duration of ice cover, which directly influences many lake mixing regimes, is expected to decrease on average by $29 \pm 8$ days by 2080–2100 in seasonally ice-covered lakes globally under future climate change scenarios (*Woolway & Merchant, 2019*). Changes in ice cover can fundamentally alter lake ecosystems, influencing lake hydrodynamics, oxygen availability, and organismal habitat (*Salonen et al., 2009*; *Hampton et al., 2017*; *Flaim et al., 2020*). Overall, given that different metrics of lake thermal structure will likely have varied future responses to climate change, developing projections for multiple thermal metrics—as well as quantifying their associated uncertainty—is critical when considering the ecological impacts of climate change on lakes.

To better interpret and appropriately use projections of future lake thermal structure, quantifying the different sources of uncertainty associated with model projections is critical for bounding predictions of future lake ecosystem changes. While many sources of uncertainty exist, the sources most commonly quantified in environmental forecasts and projections are model driver data uncertainty (*i.e.,* uncertainty in estimates of model inputs, such as meteorological driver data), initial condition uncertainty (*i.e.,* uncertainty in the model's initial states), observational uncertainty (*i.e.,* uncertainty in the actual measurement

of the variables being modeled), parameter uncertainty (*i.e.,* uncertainty in the values of a model's parameters), and model process uncertainty (*i.e.,* uncertainty in the modeled representation of complex ecosystem processes within a model; *Dietze, 2017*; *Her et al., 2019*; *Heilman et al., 2022*; *Golub et al., 2022*). Process model selection uncertainty (*i.e.,* the uncertainty in having multiple process models simulate the same target variable) and driver model selection uncertainty (*i.e.,* the uncertainty in having multiple driver models simulate the same variable that becomes input to the process model) are largely overlooked, and, to the best of our knowledge, have never been compared with one another in lake thermal projection studies (*Moore et al., 2021*; *Feldbauer et al., 2022*). While interactions across some of these uncertainty types have been examined in projection studies in other ecosystems (*e.g.,* *Wada et al., 2013*; *Hoan, Khoi & Nhi, 2020*; *Heilman et al., 2022*), the relative importance of different sources of uncertainty remains largely unexplored in lake projections.

One commonly-used method to estimate uncertainty in projections is ensemble modeling (*Parker, 2011*). For longer-term lake projections specifically, this approach entails using one or more climate scenarios, fed into one or more climate models that generate weather data, which are then used as inputs (*i.e.,* drivers) to one or more lake models to produce an ensemble projection of different lake thermal metrics. Comparing the output of ensemble members provides a more realistic representation of the diverse spread of model outcomes, as well as an opportunity to examine the effects of interactions between climate models and lake models on lake thermal projections. Importantly, by predicting lake thermal properties using multiple lake models and climate models, we can better quantify the uncertainty in future lake responses to climate change.

To date, long-term (decade to century) lake thermal projection studies have generally only quantified one or two possible sources of uncertainty in ensemble modeling. For these long-term studies, lake model selection uncertainty (*i.e.,* uncertainty derived from the decision of choosing a single lake process model from a suite of possible models), driver uncertainty, and parameter uncertainty are occasionally examined individually, with comparisons across uncertainty sources largely neglected (*Kobler & Schmid, 2019*; *Golub et al., 2022*). An exception is *Feldbauer et al. (2022)*, who found that lake model selection uncertainty was a more important contributor of uncertainty than driver data or parameter uncertainty in projections of water temperature for a German reservoir. However, they used a single climate model to produce meteorological driver data and therefore did not account for the role of climate model selection uncertainty, which limits their ability to quantify the overall and proportional contribution of both climate model and lake model selection uncertainty. Climate model selection uncertainty, a type of driver data uncertainty derived from the decision of choosing a single climate driver model from a suite of possible climate driver models, has previously been examined in projections for other ecosystems but not lakes specifically. Thus, an opportunity to expand on *Feldbauer et al. (2022)* would include analyzing both lake model selection uncertainty and climate model selection uncertainty with an ensemble approach using multiple climate scenarios, as we are unaware of any studies that analyze uncertainty across all three components of multiple climate scenarios, climate models, and lake process models.

In this study, we generated ensemble projections of future lake thermal dynamics for a dimictic lake in New Hampshire, USA (Lake Sunapee). We quantified the role of lake model selection uncertainty and climate model selection uncertainty in lake thermal projections for this seasonally ice-covered lake by using four General Circulation Models (GCMs) as inputs to five lake models. Our ensemble projections also used three climate scenarios as inputs to the GCMs, which represent a range of possible greenhouse gas radiative forcing pathways. We made projections for a set of lake thermal metrics over the next century relative to a historical baseline. Our objectives were to: (1) project future thermal structure for Lake Sunapee with quantified uncertainty across three climate scenarios up to the year 2099, (2) examine uncertainty dynamics among lake models, climate models, and their interactions between mid-century and end of century, and (3) partition the relative contributions of climate model selection uncertainty (from the four GCMs) and lake model selection uncertainty (from the five lake models) over time for each thermal metric. Altogether, our study aimed to improve our understanding of lake thermal responses to climate change and quantify the contribution of overlooked sources of uncertainty for projections of future lake dynamics.

## MATERIALS & METHODS

### Overview

To create ensemble projections of lake thermal dynamics, we used four GCMs to drive five vertical one-dimensional (1-D) hydrodynamic lake models. To explore the effects of different potential radiative forcing scenarios on future climate, we used GCM output from three representative concentration pathways (RCPs). Each lake model was calibrated with ten years of historical water temperature data (2005–2015) using standard methods (described below in 'Materials & Methods: Lake Model Calibration and Validation') to a minimum Root Mean Square Error (RMSE) for six lake thermal metrics (described below in 'Materials & Methods: Thermal Metrics'). The relative performance of each model following calibration was evaluated using five years of validation data (2015–2020; Fig. 1.1). After calibration and validation, projections were run from 1938 to 2099, including a spin-up period (1938–1974) to minimize the impact of initial conditions on the simulations, a historical period used to calculate a historical baseline (1975–2005), and a future climate projection period (2006–2099; Fig. 1.2). Anomalies were calculated for all future projections based on the difference between the historical and projection periods to determine the change in each lake thermal metric. We grouped projection output by lake model and GCM to determine model interactions over mid- and end-century. Lastly, we partitioned the relative contributions of lake model selection uncertainty and GCM climate model selection uncertainty over time for all thermal metrics across the paired RCP, GCM, and lake model combinations (Fig. 1.3). We note that our study calculated lake model and climate model selection uncertainty using an ''ensemble of opportunity'' (following *Tebaldi & Knutti, 2007*), in that we quantified the uncertainty in our unique selection of lake models and climate models from a set of available models, which was not exhaustive of all possible ways the system could be modeled (*Parker, 2011*). This approach

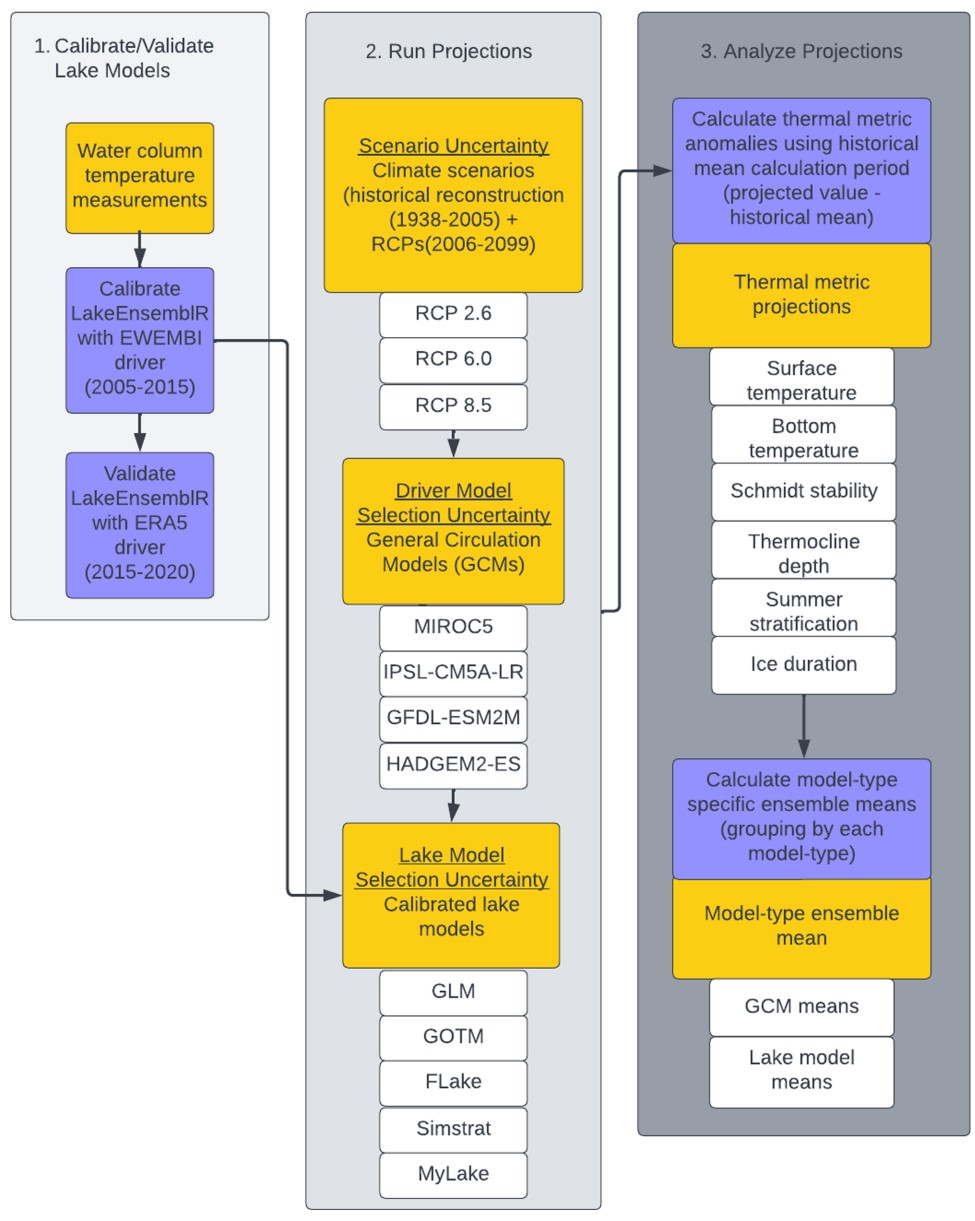

**Figure 1** **Conceptual workflow representing the methodology used to produce lake thermal projections.** Methodological workflow explaining the full projection process starting from Step 1, using observational data to calibrate and validate lake models; Step 2, using a scaffolded approach to create an ensemble of $n = 60$ model projections which incorporates climate model selection uncertainty (*via* four GCMs), and lake model selection uncertainty (*via* 5 lake models); and Step 3, analyze the projection output by first calculating anomalies for each thermal metric using the historical mean calculation period, and then calculating model-type ensemble means (see 'Methods and Materials: Ensemble means across RCP scenarios'). Yellow boxes represent outputs or inputs in the workflow, while blue boxes represent actions.

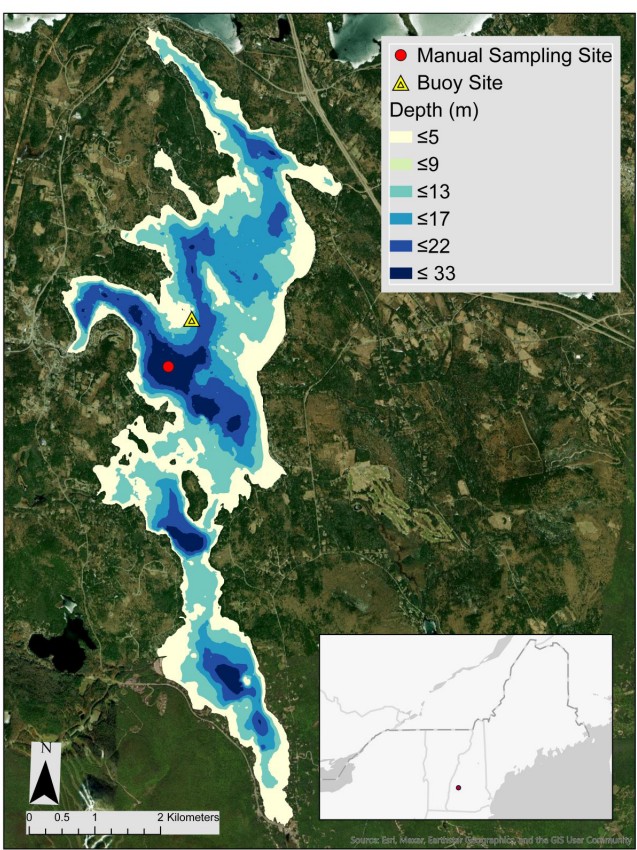

**Figure 2 Map of Lake Sunapee, NH.** Location and bathymetry of Lake Sunapee, New Hampshire, USA (43.39745°N, 72.05065°W), showing the site of the Lake Sunapee Protective Association GLEON buoy where high-frequency temperature measurements are collected on m intervals to 10 m depth. The red dot denotes a manual sampling location, where monthly water temperature profiles were measured on m intervals to 30 m depth.

means that using other lake and climate model combinations could result in novel and unique findings based on the properties of the models chosen (*Tebaldi & Knutti, 2007*).

## Study site

We generated lake thermal projections for Lake Sunapee, an oligotrophic lake located between Merrimack and Sullivan Counties in New Hampshire, USA (43.37, −72.05; Fig. 2). The lake is deep ($Z_{maximum}$ = 33 m) and dimictic, with ice cover ranging from December or January until March or April (*Bruesewitz et al., 2015*; *LSPA and Town of Sunapee, 2022*). From 2016 to 2020, the mean observed ice duration was 102 days, with a range between 55 and 128 days. Summer stratification typically occurs from mid-June to late September, with a summer thermocline depth of 6–8 m (*Carey et al., 2014*). From 1979 to present, the Lake Sunapee region has experienced an increase in observed air temperature at a rate of 0.42 °C per decade and substantial land use change in the surrounding catchment (*Cobourn et al., 2018*; *Ward et al., 2020*).

## Representative concentration pathways

To encompass a range of potential future climate scenarios, we used three representative concentration pathways (RCPs) in our study: RCP 2.6, RCP 6.0, and RCP 8.5 (Table S1). RCP scenarios were developed by the Intergovernmental Panel on Climate Change (IPCC) based on several socioeconomic factors, including land use and cover data, as well as greenhouse gas emissions (*Van Vuuren et al., 2011*; *Frieler et al., 2017*). The three RCP scenarios range from low to high climate forcing impact, with RCP 2.6 having the lowest radiative forcing level of 2.6 W/m$^2$ by the end of the century, RCP 6.0 representing a medium forcing level of 6.0 W/m$^2$ by the end of the century, and RCP 8.5 representing a high forcing level of 8.5 W/m$^2$ by the end of the century (Table S1). To date, RCP 8.5 is the best match with current trends to at least mid-century under current and stated policies (*Schwalm, Glendon & Duffy, 2020*).

## General circulation models

To represent the atmospheric conditions of future climate, we coupled each RCP scenario with four general circulation models (GCMs): MIROC5, IPSL-CM5A-LR, GFDL-ESM2M, and HADGEM2-ES (Table 1). The models were developed in four different climate research laboratories across the world, using different approaches to represent global climate processes. These models were chosen following the Inter-Sectoral Impact Model Intercomparison Project (ISIMIP), which is an international effort to better understand climate projections and their uncertainties using ensemble modeling (*Frieler et al., 2017*; *Golub et al., 2022*). ISIMIP provides a consistent modeling framework that reduces fragmentation and methodological differences across studies. Specifically, we used the ISIMIP2b lake sector inputs which consistently bias-correct GCMs on a global grid. All GCM outputs were downloaded from the ISIMIP database (available at https://www.isimip.org/gettingstarted/availability-input-data-isimip2b/), where they were bias-corrected to a 0.5° × 0.5° grid using the CDF-t (cumulative distribution function—transform) method (*Lange, 2017*) with the EartH2Observe, WFDEI and ERA-Interim data Merged and Bias-corrected for ISIMIP (EWEMBI) data (Table 1), and archived along with all model code for this analysis for reproducibility (*Wynne et al., 2023*). GCM output ranged from 1861–2005 based on historically reconstructed climatic conditions representing the industrialization period, as well as from 2006 to 2099, based on each individual RCP.

We recognize that the study could have included more GCMs under a greater number of RCP scenarios, such as *Her et al.*'s (*2019*) study using 35 GCMs to simulate watershed hydrology. However, we chose to maintain consistency across past and future studies by following the ISIMIP2b protocol, which was designed as a framework of four GCMs to understand the impact of global warming in the range of 1.5 °C to 2 °C on lakes (*Frieler et al., 2017*). These four GCMs were specifically selected because they best represent the uncertainty of future climate as shown by *Ito et al. (2020)*.

## Lake models

To represent future thermal conditions within Lake Sunapee, we coupled RCP-GCM output with an ensemble of five one-dimensional (1-D) lake models which simulate lake thermal

**Table 1** Description of General Circulation Models (GCMs) and lake models used in this study.

| Model type | Name | Abbreviation and version | Components | Reference |
|---|---|---|---|---|
| GCM | Geophysical Fluid Dynamics Laboratory Earth System Model (GFDL) with Modular Ocean Model version 4 (MOM4) component (ESM2M) | GFDL-ESM2M | Coupled carbon-climate earth system model with modular ocean model using vertical pressure layers | *Dunne et al. (2012)* |
| | Met Office Hadley Centre Earth System Model | HADGEM2-ES | Terrestrial and oceanic ecosystems; Tropospheric chemistry | *Collins et al. (2011)* |
| | Institut Pierre-Simon Laplace Climate Model 51 –Low Resolution | IPSL-CM5A-LR | Interactive carbon cycle, tropospheric and stratospheric chemistry, comprehensive representation of aerosols | *Dufresne et al. (2013)* |
| | Model for Interdisciplinary Research on Climate | MIROC5 | Atmosphere, ocean, sea ice, terrestrial | *Watanabe et al. (2010)* |
| Lake Model | General Lake Model | GLM v3.1.17 | Incorporates a Lagrangian layer structure with limited numerical diffusion of the thermocline | *Hipsey et al. (2019)* |
| | General Ocean Turbulence Model | GOTM v3.2 | One-dimensional water column model with state-of-the-art turbulence closure models | *Li et al. (2021)* |
| | Fresh-water Lake Model | FLake v1.0 | Bulk model capable of predicting vertical temperature profiles and mixing conditions on the time scale of hours to years | *Mironov (2010)* |
| | Simstrat | v2.4.1 | Turbulent and kinetic energy dissipation; Diffusive mixing | *Perroud et al. (2009)* |
| | Multi-year Lake Simulation Model | MyLake v1.2 | Daily vertical distribution of lake water temperature, density stratification, and seasonal lake-ice and snow | *Saloranta & Andersen (2007)* |

properties (Table 1). These models, which use different deterministic modeling approaches to simulate lake hydrodynamics with model-specific parameters and calculations, include the: (1) Freshwater Lake model (FLake) v1.0, which simulates lake systems using a two-layer parametric representation focusing on heat budget (*Mironov, 2010*); (2) General Lake Model (GLM) v3.1.0, which applies a Lagrangian structure to replicate mixing dynamics (*Hipsey et al., 2019*); (3) General Ocean Turbulence Model (GOTM) v3.2, which is a vertical 1-D hydrodynamic k-epsilon turbulence model (*Li et al., 2021*); (4) Multi-year Lake simulation model (MyLake) v1.2, which simulates daily vertical profiles of water temperature, seasonal ice and snow cover as well as other variables (*Saloranta & Andersen, 2007*); and (5) Simstrat v2.4.1, which is a vertical 1-D hydrodynamic model combining

a buoyancy-extended k-epsilon model with seiche parameterization originally developed by *Goudsmit et al. (2002)* (Table 1; *Moore et al., 2021*). Each model uses lake hypsography and daily meteorological input data to simulate water column temperature and ice cover on a daily timescale (see 'Materials & Methods: Input data for lake model calibration and validation' for information on input data). We used LakeEnsemblR (LER), an R package for predicting lake thermal dynamics using a suite of lake models (*Moore et al., 2021*), over our calibration, validation, and projection time periods to create an ensemble of lake thermal metrics, described below. These five models have been iteratively updated over time, however, we used the same model versions as *Moore et al.*'s (*2021*) LakeEnsemblR v1.0 release to maintain consistency throughout our study and for comparison with that earlier work.

## Thermal metrics

We chose six ecologically important thermal metrics for comparison across models: mean surface and mean bottom temperatures (hereafter, surface temperature and bottom temperature, respectively), Schimdt stability, summer thermocline depth, summer stratification duration, and total ice cover duration. All thermal metrics were calculated over the open-water season when observations were available (*i.e.,* following ice-off in April or May and preceding ice development in October or November), except for the three seasonally-based metrics: ice duration, summer thermocline depth, and summer stratification duration. Schmidt stability (the stability of a water body's thermal stratification and its resistance to mixing, in $J/m^2$) and summer thermocline depth (the depth of greatest density change in the water column due to differences in water temperature from June to August) were calculated using the package rLakeAnalyzer (*Read et al., 2014*). Surface and bottom temperatures, summer stratification duration (length of time stratified over the whole year not including inverse stratification during ice cover), and total length of ice duration were calculated using the LakeEnsemblR package (*Moore et al., 2021*). Surface temperatures were taken from a depth of 1.0 m and bottom temperatures were taken from 30.0 m.

## Input data for lake model calibration and evaluation

Observations of water temperature at Lake Sunapee from multiple data sources were used to calibrate and validate the five hydrodynamic models in LER. First, water temperature observations were collected approximately every meter from the surface to 30.0 m using manual thermal profile measurements collected approximately monthly in the summer from 1986–2021 (*Steele, Weathers & LSPA, 2021*). Because full profiles down to 30.0 m were collected <5 times each year, we also used data from a monitoring buoy deployed by the Lake Sunapee Protective Association (LSPA) in 2007, providing high-frequency (10-minute collection interval) temperature profiles every meter from the surface to 10.0 m from 2007 to present (*LSPA, Steele & Weathers, 2022a*). Because these datasets are from two different but nearby sites, we performed a Pearson correlation between available data at the two locations and found a high level of agreement ($r = 0.93$, $p < 0.0001$, Fig. S1), providing confidence in our choice to use data from the manual sampling site when available to

inform the modeling of water temperature dynamics deeper than 10 m. Observations of the date of ice-off have been collected yearly since 1869 and ice-on since 2016 (*LSPA and Town of Sunapee, 2022*). As a result, ice-off data were used during both calibration and validation to estimate the ability of the models to simulate ice-off dynamics.

Lake hypsography and meteorological data were used as lake model inputs during calibration and validation. The EWEMBI data product from ISIMIP (1979–2016; *Lange, 2019*) was used as meteorological forcing data for calibrating all LER models. EWEMBI data were used in place of locally collected meteorological data to calibrate the lake models to maintain consistency with GCM projections, which were bias-corrected by ISIMIP using EWEMBI data. Because the EWEMBI data product was revised in 2016, we used the next-generation EWEMBI data product ECMWF Reanalysis v5 (ERA5) to drive the lake models during validation. A comparison between EWEMBI and ERA5 during an overlapping time period of 1975–2016 showed very similar trends between the two data products indicating negligible influence on our lake temperature simulations (Fig. S2). All meteorological data are available at *Wynne et al. (2023)* within the "met_files_nc/EWEMBI" directory. Hypsography data are available at *Wynne et al. (2023)* within the "LER_inputs" directory and also published at *LSPA (2023)*.

Despite that Lake Sunapee has multiple inflows (*Ward et al., 2022*),  there is limited data availability at each of those sites. In addition, because of the long residence time of Lake Sunapee (3.1 years), it is likely that changes in inflow dynamics have minimal influence on thermal dynamics at the sampling site. As a result, we configured all LER models to run without inflow or outflow data, using a mass balance approach which maintained consistent water levels with observed dynamics (Fig. 3 shows simulations from the surface to 33.0 m, the maximum depth of Lake Sunapee).

## Lake model calibration and validation

All five of the lake models in LER (Table 1) were calibrated over a 10-year period (27 June 2005 to 1 January 2015) and validated over a five-year period (11 June 2015 to 1 January 2020; Fig. 1). The calibration years were chosen because they covered a wide range in annual temperature and precipitation (*LSPA, Steele & Weathers, 2022b*) and included the only continuous winter of high-frequency thermal profiles under ice (2007–2008; *Bruesewitz et al., 2015*; *Brentrup et al., 2021*). To avoid errors associated with initial conditions, a model spin-up period of 180 days was removed from the beginning of the calibration period as well as the validation period, following a similar spin-up period in another Lake Sunapee modeling study (*Farrell et al., 2020*).

To maximize the use of available data during the open-water season, calibration and validation of the lake thermal metrics (except for the three seasonally-based metrics: ice duration, summer thermocline depth, and summer stratification duration) were assessed from ~April or May until ~October or November, depending on when the buoy was deployed. Similarly, due to the low data availability of deep-water temperature profiles (~monthly during the open-water season), we calculated observed stratification duration only using observed and modeled output from the surface to 10.0 m, which encompassed typical summer thermocline depths but missed deeper temperatures. Because the data

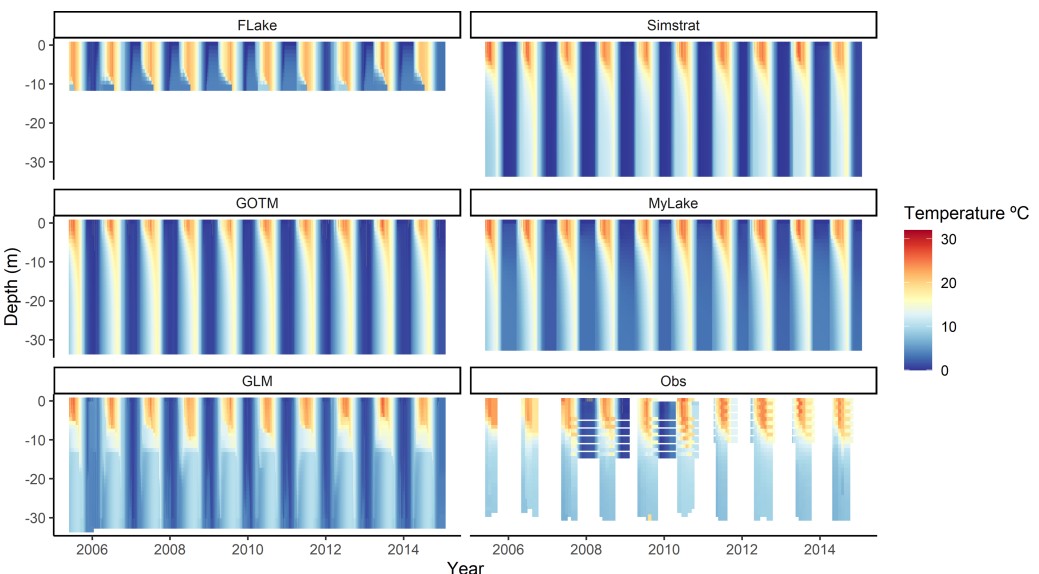

**Figure 3  Modeled and observed water column temperatures over the calibration period.** Contour plot of whole water column temperature profiles from the surface to the bottom of the lake (with the exception of FLake (surface to 11.0 m) and MyLake (surface to 32.0 m); see 'Methods: Calibration') during the calibration period (2005–2015) for each lake model (FLake, GLM, GOTM, Simstrat and MyLake). Observed water temperature data (Obs) for Lake Sunapee, NH are shown in the bottom right panel.

record for ice-on began only recently relative to the record for ice-off (*Bruesewitz et al., 2015*; *LSPA and Town of Sunapee, 2022*), we compared observations of the date of ice-off to model output during calibration and validation, while during the projection time period we calculated the total length of ice duration (between ice onset and ice-off). Despite that data availability (especially for ice-on dates) remains a major limitation for modeling lake ice dynamics, integrating the ice records we did have with process-based modeling provided an opportunity to examine changes in ice dynamics over the next century (following *Sharma et al., 2020*).

Calibration was carried out using Latin Hypercube (LHC) sampling of parameters with 500 iterations within LER, which uses upper and lower bounds for selected parameters and samples evenly within the bounded parameter space (*Mckay, Beckman & Conover, 2000*). This approach has performed well in numerous lake modeling studies (*Makler-Pick et al., 2011*; *Gal, Makler-Pick & Shachar, 2014*; *Moore et al., 2021*; *Feldbauer et al., 2022*; *Desgué-Itier et al., 2023*). The scaling factors for wind speed and shortwave radiation were calibrated for all models. However, due to structural differences among models, different model-specific parameters were calibrated for each model, which resulted in slight variation in performance among models (*Moore et al., 2021*; see Table S2 for a full list of parameter descriptions and values). Because LHC sampling calibrated all models using the same method but targeted different key parameters within each model, there was a small range of model skill across all five models, but all were within reasonable skill levels for the whole water column RMSE (~2°C; *Moore et al., 2021*). We calibrated each of the models to match either the whole water column temperature (GLM, GOTM, Simstrat, MyLake)

or the temperature to the mean water column depth (FLake). We chose to use mean water column depth for all FLake simulations as the model assumes a rectangular-shaped basin with constant mean depth, as opposed to leveraging a hypsographic curve (*Mironov, 2010*). This approach has performed well simulating thermal dynamics using the mean depth in other studies (*Woolway & Merchant, 2019*; *Feldbauer et al., 2022*; *Moore et al., 2021*).

Following calibration of the whole water column temperature, we assessed each of our six target thermal metrics (see 'Materials & Methods: Thermal Metrics') using RMSE and bias (mean error), as well as a Taylor diagram. Taylor diagrams evaluate multiple aspects of complex models by quantifying and visualizing the correlation (shown *via* straight dotted lines), centered root-mean square difference (shown *via* arced lines), and the magnitude of variability represented by standard deviation of the observations ($x$-axis) and the model output ($y$-axis; *Taylor, 2001*). We compared observations to modeled values of daily mean surface and bottom temperature, annual mean Schmidt stability, summer thermocline depth, summer stratification duration, and total ice duration. After calibration, we used the final parameters from the calibration period (2005–2015) to validate our models from 2015–2020, for which we used the same goodness-of-fit metrics (RMSE, bias). While maintaining constant parameter values prevented us from quantifying the contribution of parameter uncertainty in our lake thermal projections, other lake and hydrological modeling studies have found parameter uncertainty to contribute a small fraction of total uncertainty (*Her et al., 2019*; *Thomas et al., 2020*), and our overarching goal was to isolate the effects of climate model and lake model selection uncertainty. However, we acknowledge that parameter uncertainty is inherently linked to lake model selection uncertainty based on the parameters used in a given lake model, and may contribute to overall lake model selection uncertainty.

## Climate projections

Following calibration, lake models were run for the entire simulation period from 1938–2099. This included a spin-up period (1938–1974), historical mean calculation period (1975–2005), and a climate projection period (2006–2099; Fig. 1). From 1938–2005, the GCMs were run using a historical reconstruction, while from 2006–2099, GCMs were run using three RCP climate scenarios.

To represent deviation from historical trends, we calculated an annual anomaly from the historical baseline (1975–2005) for each thermal metric over our projection time period (2006–2099). First, we calculated the mean annual value for each thermal metric over the entire historical mean calculation period. Then, anomalies from this historical period were calculated on a daily time step within the projection period for each thermal metric and averaged to an annual value for each metric.

## Ensemble means across RCP scenarios

From our 60 unique model projections (3 RCP scenarios × 4 GCMs × 5 lake models), we calculated ensemble means aggregated for each RCP scenario to summarize the impact of each climate scenario on the six thermal metrics ('Materials & Methods: Thermal Metrics'). These ensemble means included all GCMs and lake model projections, resulting in three
ensemble means of 20 projections (4 GCMs × 5 lake models) for each of the six thermal metrics. Ensemble means were calculated annually over the entire projection time period (2006–2099). To represent uncertainty across projections, we also calculated the standard deviation across the 20 projections for each RCP scenario.

## Ensemble model interactions

Ensemble modeling incorporates multiple models which represent the same processes in different ways, resulting in distinct dynamics for each model output (*e.g.*, differences between GLM and Simstrat). Additionally, different combinations across model types (*e.g.*, a GCM and a lake model) can potentially produce unique interactions which may further influence the magnitude and direction of projections. To examine the effects of individual models as well as the interactions between model types, we calculated model-type specific ensemble means which captured variability across all GCMs for a single lake model and vice versa. We calculated these ensemble means separately for each RCP scenario.

To calculate model-type ensemble means within a single RCP scenario, projection output was grouped by uncertainty type (*e.g.*, a GCM or lake model), and the mean was calculated across an individual model within each type. For example, to compare the impact of an individual GCM (*e.g.*, GFDL-ESM2M) on all lake models, the ensemble mean was calculated across all five lake model outputs annually (and then grouped by 30-year periods) driven by GFDL-ESM2M climate data (Fig. S3). Similarly, to calculate the impact of a single lake model (*e.g.*, GOTM), the mean of all five GCM × GOTM outputs was taken (Fig. S3).

From the annual model-type means, we further grouped each model-type mean by 30-year intervals and examined distributions at mid-century (2020–2050) and end-century (2069–2099). Examining projection output over these 30-year intervals reduces climatic noise due to inter-annual variation in climate projections (*Fischer et al., 2012*). Within the model-type ensemble mean distribution (*i.e.,* the distribution of individual model outputs within a model-type; Fig. S3), we specifically looked for multimodality, which would signify a disagreement between models, and unimodality, which would signify a high level of agreement between models. We also examined the level of agreement across model-type means (*e.g.*, lake models *vs.* GCMs) by examining their respective distributions with one another. Distributions with similar means and ranges of projected values among model-types were considered more robust than those with very different distributions. Within a distribution, we identified individual lake and GCM models which resulted in outliers by examining the full time series of each model-type mean (Figs. S4–S9).

## Uncertainty partitioning and quantification

To determine the relative influence of climate model selection uncertainty and lake model selection uncertainty on total projection uncertainty, we partitioned the relative contribution of each uncertainty type for each lake thermal metric across all projections ($n = 60$ total). For each thermal metric, we calculated lake model selection uncertainty by calculating the variance across the five lake models for all GCM and RCP combinations (*Moore et al., 2021*). Further, we calculated climate model selection uncertainty by

calculating the variance across the four GCMs for all lake model and RCP combinations. Total projection uncertainty was calculated as the total standard deviation across all projection outputs, which included climate model selection uncertainty and lake model selection uncertainty. To calculate proportional variance for the two uncertainty types, we divided the variance of each respective uncertainty type by their combined total uncertainty. Our method follows the uncertainty quantification and partitioning approaches used in several studies across the field of environmental forecasting and projections (*e.g.*, *Diniz-Filho et al., 2009*; *Buisson et al., 2010*; *Dietze, 2017*; *Thomas et al., 2020*; *Woelmer et al., 2022*). In the results below, we primarily report our findings of RCP 8.5 for the uncertainty analyses because it has been shown to be the most accurate RCP relative to recent $CO_2$ concentrations and global temperatures (*Schwalm, Glendon & Duffy, 2020*); all other RCP scenario results are presented in the SI (Figs. S4–S7, S10–S13).

All analyses were conducted in R v.4.0.2 (*R Core Team, 2020*). All data used in this study are published (manual water temperature: *Steele, Weathers & LSPA, 2021*; buoy water temperature: *LSPA, Steele & Weathers, 2022a*; ice-off: *LSPA and Town of Sunapee, 2022*; lake hypsography: *LSPA, 2023*). All code to run the analyses, including the downloaded GCM × RCP meteorological driver inputs and model initialization files are available on Zenodo (*Wynne et al., 2023*). All projection output to recreate the analyses is archived on Zenodo (*Wynne et al., 2022*).

# RESULTS

## Lake models reproduce observed thermal dynamics

Throughout the calibration period, all lake models reproduced observed Lake Sunapee thermal structure and stratification patterns (Fig. 3). Four out of five of the lake models reproduced modeled whole water column temperature with a root mean square error (RMSE) of <2 °C for whole water column temperatures, except for FLake with an RMSE of 2.23 °C (Table S3). The ensemble mean across all lake models performed as well or better for multiple metrics during calibration and validation compared to the best performing individual model, with a whole water column temperature RMSE of 1.29 °C and bias of −0.15 °C during calibration, and an RMSE of 1.69 °C and bias of −0.04 °C during validation (Tables S2 and S3).

Within the water column, model performance varied with depth. All models reproduced surface temperature observations well, with high correlation between modeled output and observations ($r = 0.99$, Fig. 4A), and RMSE <1.51 °C (Table S3). In contrast, the models reproduced observed bottom temperature with less skill, with RMSE ranging from 2.52–4.69 °C (Table S3) and $r < 0.4$ (Fig. 4B). However, we note that these high error metrics are likely due to much lower frequency of data availability at bottom temperatures (30.0 m; $n = 20$) compared to surface temperatures (1.0 m; $n = 900$). Because FLake only simulates the surface layer of lakes (see 'Materials & Methods: Calibration'), output from this model was removed from the bottom water temperature calibration and validation calculations. Similarly, MyLake only simulates from the surface to 1.0 m above the bottom (here, 32.0 m) and thus its output was analyzed accordingly across these depths.

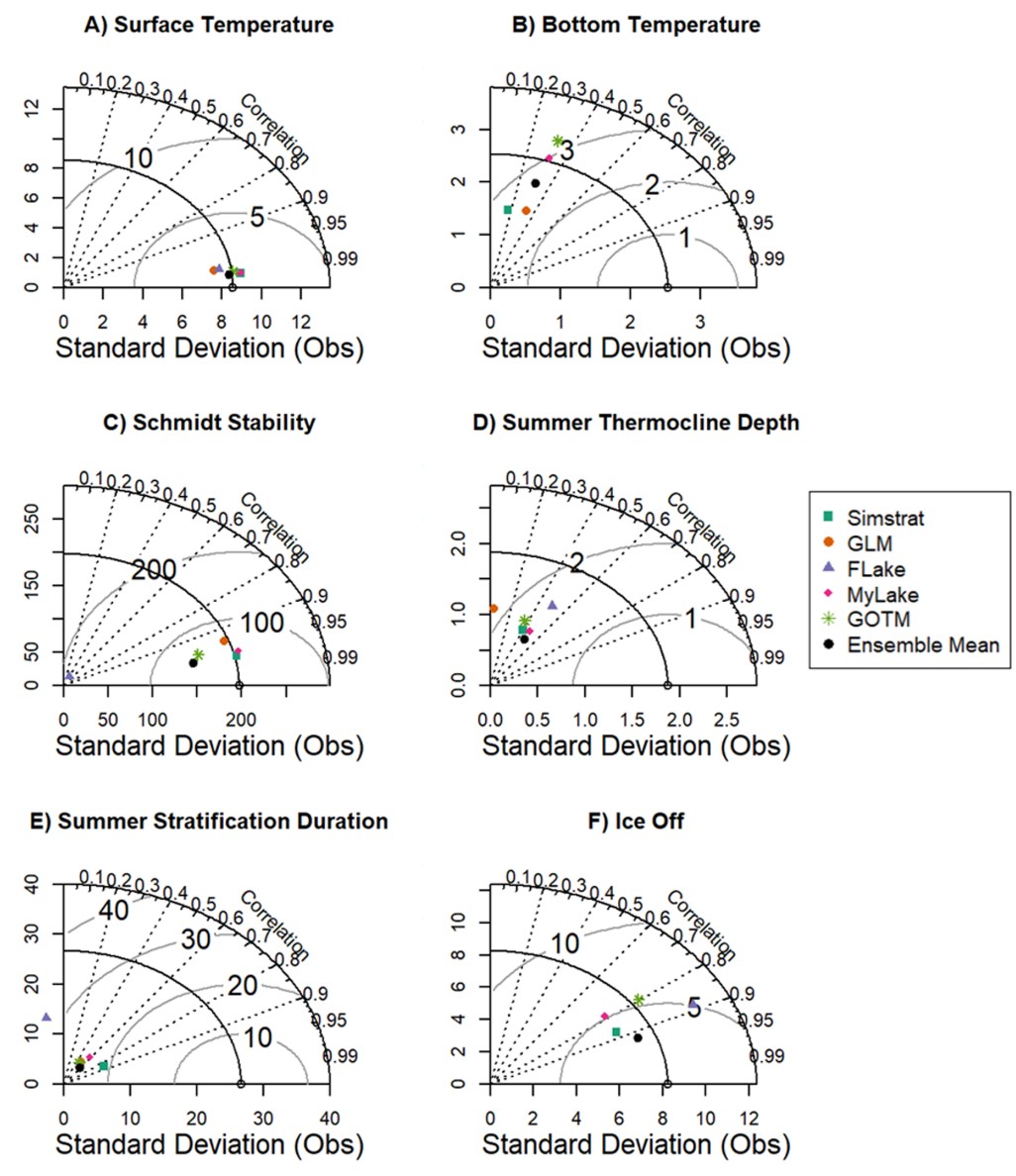

**Figure 4** **Taylor diagrams showing model performance.** Taylor diagrams showing the standard deviation of observations ($x$-axis) and modeled thermal metrics ($y$-axis) for Lake Sunapee, NH, USA during 2005–2015. The correlation between observations and modeled output is shown as dotted lines, and root mean squared error (RMSE) between observed and modeled standard deviation is shown as arcs. Correlation ($r$) represents the linear trend between observed and modeled data. Each colored point represents a specific lake model compared with observed data. Thermal metrics include (A) summer surface temperature mean, (B) summer bottom temperature mean, (C) Schmidt stability, (D) summer thermocline depth, (E) stratification duration, and (F) date of ice-off. Summer is defined as June–August. Values to the left of the $y$-axis (*e.g.*, panel E) represent negative correlation.

All models reproduced Schmidt stability well but were less skillful at capturing summer thermocline depth and summer stratification duration. Modeled Schmidt stability from FLake exhibited a high correlation to the observed data ($r > 0.95$, Fig. 4C), but a low
standard deviation compared to the observations, indicating that the variability in modeled output was smaller than observed variation (Fig. 4C). Overall, Simstrat reproduced Schmidt stability better than the other models (Fig. 4C; RMSE = 46.36 J/m², Table S3). Summer thermocline depth was reproduced relatively poorly by all lake models, with $r < 0.50$ (Fig. 4D), but with an overall ensemble RMSE of 1.65 m during calibration and 2.18 m during validation (Table S3). The worst performing model for thermocline depth was GLM, with $r < 0.10$ (Fig. 4D). Summer stratification duration was also poorly reproduced, similar to thermocline depth, with an RMSE among models of 33–75 days and overall correlation of $r < 0.40$ (Table S3, Fig. 4E). However, these poor goodness-of-fit values were likely due to the low temporal resolution of the observed data during early spring and late fall, which limited our ability to capture stratification dynamics in some years.

The day of ice-off was captured well by Simstrat, MyLake and FLake, with high goodness-of-fit metrics across these models ($r > 0.8$, RMSE ~5 days) and similar standard deviation in comparison to the observed data (Table S3, Fig. 4F). In contrast, GOTM did not perform well in predicting the day of ice-off, with a higher RMSE (55 days; Table S3) and bias (55 days; Table S4) than all other models, skewing the ensemble mean RMSE to 13 days during calibration (Table S3). The version of GLM (v3.1.0) in LakeEnsemblR (v1.0) did not simulate ice and thus was not included in the ice goodness-of-fit calculations or projections.

## Warmer, more summer-stratified projections of Lake Sunapee's future

Our projections show that all six thermal metrics of Lake Sunapee will change substantially in response to climate change over the next century (Fig. 5). Surface temperature is projected to increase by 1–5 °C above historical conditions by the end of this century (Fig. 5A). Similarly, bottom temperature is also projected to increase, but to a lesser extent (1–2 °C; Fig. 5B). Metrics of stratification indicate a longer and stronger summer stratification period within the lake annually, with the duration of summer stratification increasing by 10–60 days (Fig. 5E). In addition, the strength of thermal stratification, Schmidt stability, is projected to increase by 20–100 J/m² (Fig. 5C). In contrast, total ice duration is projected to decrease by 20–75 days (Fig. 5F). Interestingly, thermocline depth is projected to stay the same over the course of the century, but with increased variability by the end of the century (Fig. 5D).

As a result of longer summer stratification and less ice cover, Lake Sunapee's mixing dynamics will be altered, with up to 63 additional days spent stratified in the summer and up to 75 fewer days spent inversely stratified due to decreasing ice cover in the winter (Fig. 5). The magnitude of anomalies for each metric were largely driven by RCP scenario, with the smallest ranges projected by RCP 2.6, followed by RCP 6.0, and the highest ranges projected by RCP 8.5 (Fig. 5), which largely followed expected patterns corresponding to the magnitude of climate change associated with each RCP scenario. For example, under RCP 2.6, which includes reduced carbon emissions by mid-century (*Van Vuuren et al., 2011*), anomalies decreased from mid- to end-century, with lower predicted temperatures, less change in stratification duration, and lower Schmidt stability values at end-century compared to mid-century, in line with the socioeconomic trajectory of this scenario.

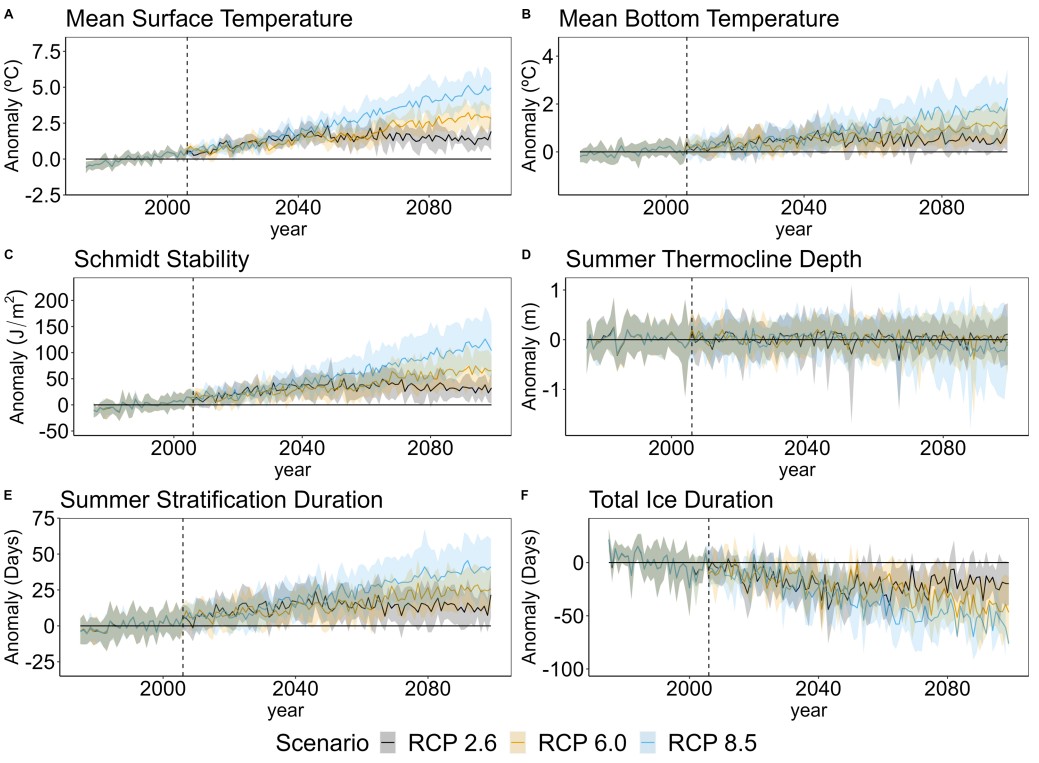

**Figure 5 Lake thermal projections through end-century.** Projected anomalies for (A) mean summer surface temperature (surface temperature), (B) mean summer bottom temperature (bottom temperature), (C) Schmidt stability, (D) thermocline depth, (E) summer stratification duration, and (F) total ice duration from 2006–2099. The vertical dashed line represents the beginning of the projection time period, with the left of the dashed line representing the historical mean calculation period on which anomalies were based (1975–2005). Each solid line represents the ensemble mean of the lake models under RCP 2.6, 6.0, or 8.5 and each shaded area around the solid lines represents total projection uncertainty under RCP 2.6, 6.0, or 8.5.

## Interactions between models from mid-century and end-century projections show model disagreement for lake bottom temperature and ice cover

From mid- to end-century, distributions of model-type means (ensemble means grouped by climate or lake models) varied with thermal metric (Fig. 6) as well as RCP scenario (Figs. S10–S11). Surface temperature had a wider climate model distribution for mid- and end-century than lake model distribution, indicating a wider range of possible values due to differences across climate models (Fig. 6A). However, within a model-type both climate and lake model distributions were unimodal, indicating a high degree of model agreement for surface temperature projections across models. In contrast, mid- and end-century bottom temperature projections showed a bimodal response due to differences across lake models but not across climate models (Fig. 6B). The bimodality was primarily due to GLM performance, in which GLM projected minimal changes in bottom temperatures (<1 °C) under all climate models (Fig. S8B), as opposed to other lake models, which projected an increase of 2–3 °C.

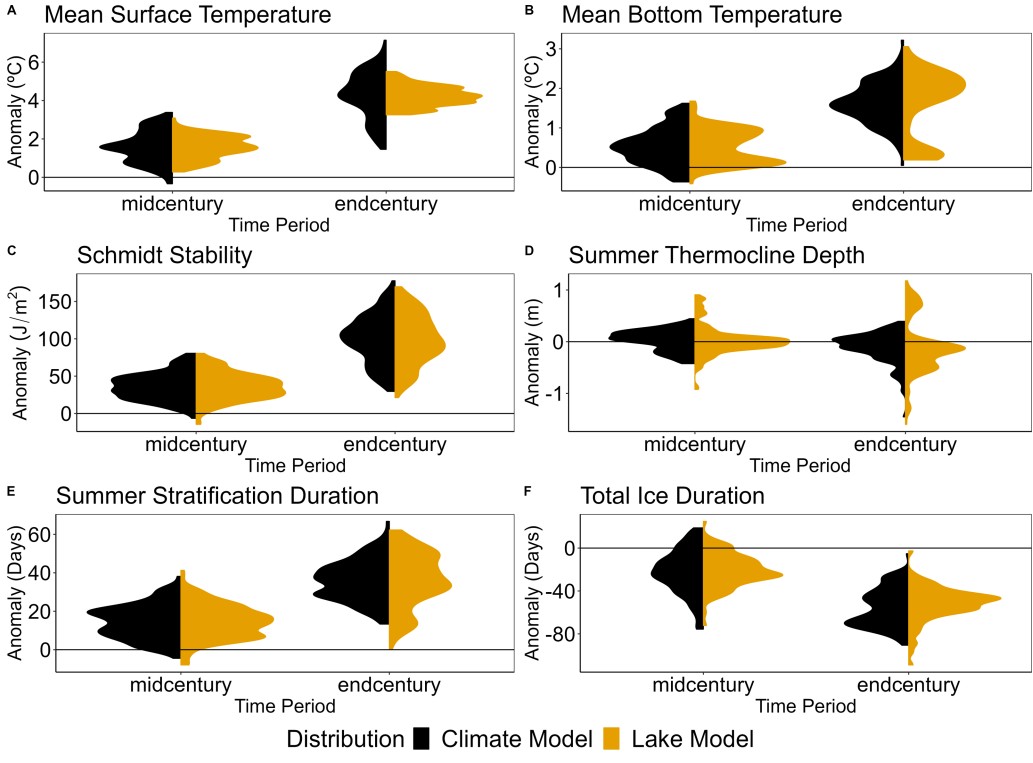

**Figure 6** **Model-type distributions for mid- and end-century across lake model and climate model uncertainty.** Distributions of the model-type mean anomalies of the climate models and lake models under RCP 8.5. These distributions were calculated for (A) mean summer surface temperature (surface temperature), (B) mean summer bottom temperature (bottom temperature), (C) Schmidt stability, (D) thermocline depth, (E) summer stratification duration, and (F) total ice duration during mid-century (2020–2050) and end-century (2069–2099).

In contrast to bottom temperature, Schmidt stability had a high degree of agreement between lake models and climate models, as shown by similar distributions between lake and climate models in mid-century and end-century (Fig. 6C). While distributions of thermocline depth anomalies were centered around zero, indicating minimal change at both mid-century and end-century, there was a large spread in the distribution of lake models as compared to GCMs by end-century, showing a wide range of disagreement in future thermocline depth due to differences in lake models (Fig. 6D, Fig. S8D). Similar to bottom temperature, this was primarily driven by GLM output, which projected a larger increase in thermocline depth (*i.e.,* negative anomaly) than other lake models (Fig. S8). FLake also contributed to the disagreement in projections of thermocline depth, as the only model which projected a decrease in thermocline depth (*i.e.,* positive anomaly). However, this result should be interpreted with caution due to GLM's poor fit in simulating thermocline depth during the calibration and validation period (Fig. 4, Table S2) as well as FLake's characteristic of only simulating water temperature to the mean water column depth and not the whole water column (see 'Materials & Methods: Calibration').
The model-type distributions showed model disagreement within and across model-types. Summer stratification duration had a bimodal distribution across lake models, particularly by end-century (Fig. 6E), driven by a lower projected anomaly from FLake (Fig. S8E). Thermocline depth showed a mostly unimodal pattern across climate models, but had a skewed, multimodal distribution across lake models due to disagreement in lake models driven by FLake and GLM (Fig. 6D, Fig. S8). Lastly, total ice duration was generally unimodal for both climate and lake models, indicating agreement within model-type (Fig. 6F). However, for the end-century projections, the lake model distribution was centered around −40 (*i.e.,* 40 fewer days of ice) while the climate model distribution was centered around −75 (*i.e.,* 75 fewer days of ice). The difference in these distributions suggest a greater degree of ice loss due to variation in climate models than lake models.

## The dominant source of uncertainty varied over time and thermal metric

The relative proportion of uncertainty due to climate and lake models varied among thermal metrics and over time for RCP 8.5 scenarios (Fig. 7). Uncertainty in surface temperature was consistently dominated by climate model selection uncertainty (>80%) throughout the entire projection period (Fig. 7A). In contrast, bottom temperature was dominated by climate model selection uncertainty up until mid-century, after which ~75% of uncertainty was due to lake model selection uncertainty (Fig. 7B). Uncertainty in Schmidt stability was dominated by climate model selection uncertainty until mid-century (~75%), after which lake model and climate model selection uncertainty contributed equally (Fig. 7C). Uncertainty in thermocline depth was evenly split by lake model and climate model selection uncertainty at the beginning of the projection period, but lake model selection uncertainty increased over time to an overall proportion of >75% by the end of the century (Fig. 7D). Total stratification duration was initially dominated by climate model selection uncertainty until mid-century (~75%), when lake model selection uncertainty became the primary source (~75%; Fig. 7E). Lastly, uncertainty in total ice duration was dominated by climate model selection uncertainty over the entire projection period (60–75%; Fig. 7F).

In all metrics but surface temperature, the proportional contribution of lake model selection uncertainty increased over time (Fig. 7). In contrast, no metrics exhibited a shift from lake model dominance to climate model dominance in proportional uncertainty over time. Specifically, for bottom temperature and total stratification duration, the dominant source of uncertainty switched from climate model to lake model mid-century (Figs. 7B and 7E). For surface temperature and total ice duration, climate model selection remained the dominant source of uncertainty throughout the projection time period (Figs. 7A and 7F). Interestingly, lake model selection uncertainty in surface temperature, Schmidt stability, thermocline depth, and total stratification duration did not increase at a constant rate and increased more quickly in the beginning of the projection period up to mid-century (Fig. 7).

The dominant source of uncertainty varied among RCP scenarios for some thermal metrics (Figs. S12 and S13). Under RCP 2.6, climate model selection uncertainty was the dominant source of uncertainty for all thermal metrics but thermocline depth (Fig.

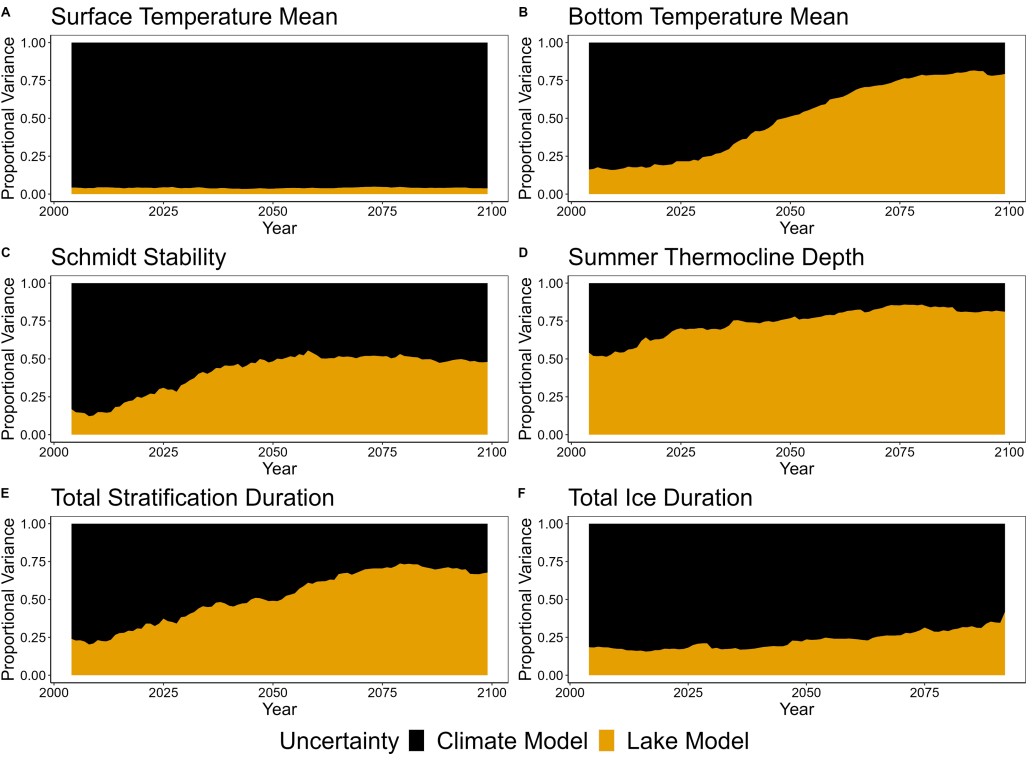

**Figure 7** **Proportional variance of the contribution of climate model selection uncertainty (black) and lake model selection uncertainty (orange) to total uncertainty.** (A) Surface temperature mean (surface temperature), (B) bottom temperature mean (bottom temperature), (C) Schmidt stability, (D) thermocline depth, (E) total stratification duration, and (F) total ice duration from 2006–2099. Proportional variance was calculated for all models using the RCP 8.5 climate scenario.

S12). For bottom temperature, summer stratification duration, and Schmidt stability, the relative contribution of lake model selection uncertainty increased up to mid-century and then began decreasing towards end-century. Under RCP 6.0, uncertainty dynamics were similar to RCP 8.5, in which climate model selection uncertainty was the dominant source for surface temperature and total ice duration, while lake model selection uncertainty was the dominant source for bottom temperature, thermocline depth, and total stratification duration by end-century (Fig. S13). Schmidt stability had similar contributions between GLM and lake model selection uncertainty for both RCP 6.0 and RCP 8.5 (Fig. 7C, Fig. S13C). For the metrics dominated by lake model selection uncertainty under RCP 6.0, uncertainty continued to increase to 2099 as opposed to leveling off (Figs. S13B, S13C and S13E), similar to RCP 8.5 (Figs. 7B, 7C and 7E), which exhibited increased uncertainty earlier in the projection period.

# DISCUSSION

## Overview

Decision-makers must have access to robust model projections with quantified uncertainties to prepare for and mitigate the effects of climate change on lakes.

Consequently, identifying the dominant sources of uncertainty (*e.g.*, lake models, climate models) and their interactions are critical for both improving the accuracy and interpretation of projections, as well as building more robust coupled model frameworks. This study approaches these challenges by using ensemble modeling across multiple climate models, lake models, and socioeconomically-driven climate scenarios. Coupling ensembles of multiple model-types increases the robustness of our projections, as previous lake modeling studies generally have only used just one climate model, one lake model, and/or one climate scenario (*e.g.*, *Her et al., 2019*; *Golub et al., 2022*; *Feldbauer et al., 2022*).

Across our ensemble projections, the majority of the thermal changes projected for Lake Sunapee were in line with previous studies, which predict warmer, longer summer stratification periods and less winter ice cover. While most models in our ensemble projections agreed on the projected changes in thermal metrics, some models exhibited entirely different trajectories from the ensemble. For example, GLM projected no change in bottom temperature, compared to the overall 1–2 °C increase in this metric projected by FLake, MyLake, Simstrat, and GOTM under RCP 8.5 (Fig. S8). Lastly, we found that the dominant source of uncertainty for each thermal metric was sensitive to depth (surface-level metrics were more sensitive to variation in the climate model), time (the dominant source of uncertainty changed mid-century for most metrics), and climate scenario (variation among lake models increased more quickly into the future under more severe RCPs). Below, we explore further our uncertainty findings, as well as the projected changes to thermal metrics in Lake Sunapee.

## Dominant uncertainties vary for each thermal stratification metric: depth matters

Thermal metrics for the lake surface were affected more by variation across the climate models used as inputs, while metrics within the water column were affected more by variation across lake models (Fig. 7). Specifically, surface temperatures and ice duration were dominated by climate model selection uncertainty. In contrast, the dominant source of uncertainty for bottom temperature, thermocline depth, and stratification duration was lake model selection. Schmidt stability, which incorporates full water column dynamics, was equally dominated by climate model and lake model selection uncertainty by end-century.

Our findings are likely due to the direct influence of atmospheric conditions on surface temperature and ice cover, as opposed to hypolimnetic or whole-water column metrics (*e.g.*, bottom temperatures, stratification metrics), which may be more sensitive to within-lake thermal stratification and mixing processes (*Kraemer et al., 2015*). While few studies exist which separately partition the relative contributions of different sources of uncertainty in predicted surface *vs.* bottom lake hydrodynamics, it has been previously shown that surface water thermal dynamics, including ice cover, are primarily driven by atmospheric forcing (*Livingstone & Padisák, 2007*; *Sharma, Walker & Jackson, 2008*; *Piccolroaz, Toffolon & Majone, 2013*) and that bottom waters are more weakly correlated to changing air temperatures (*Butcher et al., 2015*). Metrics of thermal stratification (*e.g.*, Schmidt stability, thermocline depth) are driven by a suite of processes that were simulated differently in all of the five hydrodynamic lake models, such as sediment-water interactions, inflow

dynamics, light transparency, solutes, and other processes (*Saloranta & Andersen, 2007*; *Perroud et al., 2009*; *Mironov, 2010*; *Hipsey et al., 2019*; *Li et al., 2021*), which may explain why these metrics were more sensitive to lake model selection uncertainty than climate model selection uncertainty.

Dominant sources of uncertainty in the lake metrics also changed over time into the future (Figs. S5, S7 and S9), with a pattern of increasing spread across both GCMs and lake models towards end-century. Other studies have also documented that uncertainty is not static through time (*Bonan, Lehner & Holland, 2021*; *Heilman et al., 2022*; *Schwarzwald & Lenssen, 2022*; *Woelmer et al., 2022*). In our projections, all metrics were dominated by climate model selection uncertainty at the beginning of the century, but the relative importance of lake model selection uncertainty increased over time for all metrics except surface temperature and ice duration (Fig. 7). The consistent dominance of climate model uncertainty for surface temperatures and ice dynamics is expected given that they are tightly coupled to atmospheric conditions. In contrast, increases in the relative importance of lake model selection uncertainty over the projection period for other metrics is likely due to the differing ways that each lake model simulates the effects of atmospheric conditions through the water column, leading to increases in lake model selection uncertainty. Despite each lake model having a similar performance at the beginning of the projection period, the differing sensitivity across lake models to warming temperatures over time led to a larger spread than caused by differences in climate models alone. Overall, changing uncertainty dynamics over time may indicate how different sources of uncertainty (*e.g.*, climate model selection or lake model selection) propagate through time, having important implications for the projection horizon at which each of these sources dominate.

## Comparisons of uncertainty dynamics across time scales: different contributions of uncertainty between short- and long-term predictions

Our uncertainty findings add context to the few studies that quantify similar sources of uncertainty in lake thermal projections and forecasts. In their long-term thermal projection study using the LakeEnsemblR framework for a German reservoir, *Feldbauer et al. (2022)* found that lake model selection uncertainty was as high or higher than all other quantified uncertainties (which included a single type of meteorology driver data uncertainty, parameter uncertainty, and their interactions) for surface (3.0 m) and bottom (25.0 m) water temperatures, summer stratification duration, and total ice duration. By quantifying driver selection uncertainty across multiple climate models, as opposed to one, our study builds upon this work to identify the importance of driver model selection uncertainty relative to other sources. Indeed, we found that climate model selection uncertainty was the dominant source of uncertainty in surface-level metrics such as water temperature and total ice duration (Figs. 7A and 7F), highlighting the importance of quantifying uncertainty across multiple climate models for future lake studies. Lastly, we note that *Feldbauer et al. (2022)* and our study used different approaches to quantify uncertainty, emphasizing the need for consistent approaches for uncertainty quantification to allow robust comparisons across lakes and analyses.

Comparing uncertainty dynamics across long-term lake projections and short-term forecasts can provide valuable insight into predictability of lake thermal dynamics across time scales. In their near-term forecasting study, *Thomas et al. (2020)* found that 16-day surface temperature forecasts were dominated by meteorological driver and downscaling uncertainty, while process uncertainty dominated bottom water forecasts almost entirely during the summer stratified period, similar to our findings. However, during the fall mixed period, process and driver uncertainty were nearly equal across surface and bottom water forecasts, suggesting that mixing dynamics are a key factor in the dominant source of uncertainty in lake thermal predictions (*Thomas et al., 2020*). Due to limited data availability for Lake Sunapee, we did not quantify changes during the full duration of the mixed period, motivating the need for future work to examine uncertainty dynamics in long-term projections of lakes which have year-round high-frequency data.

Overall, short-term lake thermal forecasts may have different uncertainty dynamics than long-term climate change projections, especially under RCP 8.5. For example, *Thomas et al. (2020)* found that driver uncertainty increased more than process uncertainty with time into the future across their 16-day forecast horizon. Across full century projections, we found the opposite effect, with the contribution of process uncertainty increasing to dominate or nearly dominate total uncertainty by the end of the century for bottom temperature, Schmidt stability, thermocline depth, and stratification duration under RCP 8.5. It is possible these differences are due to the forecast or projection horizon, or the time into the future being predicted, with shorter horizons being more sensitive to driver uncertainty and longer horizons more sensitive to process uncertainty (following *Adler, White & Cortez, 2020*). Altogether, this result emphasizes the importance of partitioning total uncertainty into individual contributions for comparing uncertainty dynamics across forecasts and projection horizons.

## Uncertainty findings inform future lake projections: surface- and bottom-level thermal metrics require different uncertainty quantification approaches

Our findings support the importance of using multiple lake models to inform uncertainty dynamics and improve overall projection performance. However, running multiple lake models and climate models poses computational and logistical challenges, necessitating guidelines for how to prioritize conducting multi-model ensembles. For studies focused on estimating primarily surface water variables (*e.g.*, analyses that use satellite data for lake thermal modeling), we suggest focusing on quantifying meteorological driver uncertainty, which can be done by including numerous GCMs (*Her et al., 2019*) or other relevant meteorological driver models. In contrast, in studies focusing on simulating whole water column stratification metrics, we suggest focusing on quantifying and reducing overall lake model uncertainty. Ensemble approaches which incorporate multiple process models can help better quantify process uncertainty (*i.e.,* lake model uncertainty) by incorporating multiple different structural representations of the lake process of interest (*Dietze, 2017*), often with improved prediction (*e.g.*, *Lynch et al., 2012*; *Beger, Dorff & Ward, 2014*; *Trolle et al., 2014*; *Scher & Messori, 2021*; *Sharma et al., 2021a*). Additionally, by both incorporating

more process models and statistically weighting the overall ensemble mean based on historical model performance, overall process uncertainty may be reduced (*Raftery et al., 2005*; *Spence et al., 2018*). Lastly, targeted data collection and simulation approaches which provide insight on the underlying hydrodynamic mechanisms can improve process representation in models and ultimately reduce process uncertainty (*Dietze, 2017*). The guidelines we suggest here were developed from our work in dimictic Lake Sunapee, and will be strengthened by additional studies in other lakes, which will test the robustness of these uncertainty contributions across lake and reservoir ecosystems (*e.g.*, shallow lakes, tropical systems, reservoirs).

## Importance of ensemble means for lake modeling

Over the calibration and validation period, we found that the ensemble mean across lake models was frequently the highest performing simulation relative to observations (Table S3). This finding is not novel to lake modeling alone, and has been well-documented across other ensemble modeling studies, *e.g.*, in meteorological forecasts (*Scher & Messori, 2021*), epidemic forecasts (*Sharma et al., 2021a*), ecological forecasts (*Lynch et al., 2012*), and political forecasts (*Beger, Dorff & Ward, 2014*). Within aquatic ecosystems, the ensemble mean of phytoplankton projections from three models was the best predictor of phytoplankton in a temperate lake relative to any individual model (*Trolle et al., 2014*). While modeling two lakes using the LakeEnsemblR framework, *Moore et al. (2021)* found that the ensemble mean of water temperature frequently outperformed all other models, providing support for using an ensemble of models to predict lake thermal structure.

While ensemble modeling often has the benefit of improving model performance, ensembles with models that perform very poorly relative to others may have adverse effects on the overall ensemble mean. For example, we observed individual models that negatively skewed the ensemble mean in comparison to other lake models (*e.g.*, poor performance of GOTM relative to other models in predicting ice-off, Table S3). As a result, the ensemble mean for ice-off during calibration performed worse than individual model performance of FLake, Simstrat, and MyLake. We also found differential model behavior within our projections which affected the overall projection mean. For example, GLM's bottom temperature projections showed little to no change under all RCP scenarios, while all other lake models showed an increase (Figs. S4B, S6B and S8B). As a result, GLM skewed the total ensemble mean of bottom temperatures to be colder for all RCP scenarios (Fig. 5). Overall, in most cases the ensemble mean was the best performer, but it is important to assess all of the individual models' performance when interpreting ensemble mean projections, and potentially weight ensemble means based on historical performance (*e.g.*, *Raftery et al., 2005*). Because each individual modeling case is likely to be different, with different models performing better or worse depending on lake characteristics, site-specific variability in environmental drivers or lake response variables, and in the way various processes are simulated by each lake model, we suggest that deciding how best to aggregate model results or perform statistical model-weighting should be done on a case-by-case basis.

## RCP scenario intensity impacts uncertainty dynamics

Our results suggest that changes in proportional uncertainty over time may be directly related to the magnitude of the RCP scenario, especially for bottom temperature, Schmidt stability, and summer stratification duration. Specifically, we found that for RCP 2.6, lake model selection uncertainty increased until mid-century, but then decreased towards end-century for bottom temperature, Schmidt stability, and total stratification duration (Fig. S12). In contrast, lake model selection uncertainty in RCP 6.0 exhibited a continuous increase throughout the projection period for the same metrics (Fig. S13). Further, RCP 8.5 exhibited a faster, nonlinear increase and a higher overall proportion of lake model selection uncertainty (Fig. 7) than RCP 2.6 or RCP 6.0. A possible explanation is that as the magnitude and variability of meteorological drivers changes from current and historical conditions (*e.g.*, increased air temperatures or more variable rainfall) used to calibrate each model, uncertainty around their projected values increases. Altogether, lower atmospheric forcing values were associated with lower lake model uncertainty and higher climate model uncertainty. These findings remain largely unexplored and require further research across multiple RCP uncertainty analyses.

## Future projections of Lake Sunapee's thermal structure

Our projections of Lake Sunapee generally align with (or show slightly greater warming than) global lake projection studies, which also examine thermal metrics across the next century. Stratification projections are similar, with our study predicting 10–60 more days of summer stratification duration by end-century (2099), compared to 10–35 days of more stratified conditions on average projected for lakes globally (*Woolway et al., 2021*; *Woolway, Sharma & Smol, 2022*). Similarly, our summer mean surface water temperature change projected for Lake Sunapee was higher (range of 1–5 °C) compared to the global surface temperature projection of 1–4 °C by end-century (*Woolway, Sharma & Smol, 2022*). Changes in lake bottom temperatures show less consistency across studies, with some lake bottom waters found to be warming and others cooling (*Pilla et al., 2020*). Based on our projections, Lake Sunapee's bottom waters are likely to warm by 1–2 °C during summer months. Lake Sunapee is projected to lose more ice than the global average (10–40 day loss), with a 20–75 day loss of winter ice cover by end-century (*Woolway, Sharma & Smol, 2022*). However, due to Lake Sunapee's historical pattern of ice duration, which lasts from December or January until March or April (*LSPA and Town of Sunapee, 2022*), Lake Sunapee is unlikely to be one of ~5,700 lakes at risk of completely losing ice cover this century (*Sharma et al., 2021b*).

Interestingly, our projections found no definitive change in thermocline depth under any RCP scenario. Across other lake thermal projection studies, thermocline depth also showed a variable response to climate change scenarios, with some studies reporting up to a 0.49 m shallower thermocline under RCP 6.0 (*Ayala, Moras & Pierson, 2020*), while others report a slight deepening of thermocline depth in response to multiple RCP scenarios (*Prats et al., 2018*; *Barbosa et al., 2021*), indicating uncertainty in the directionality of this metric. These inconsistent thermocline depth responses to climate change motivate the need for

future studies which examine changes in metrics which integrate thermal dynamics from the whole water column.

The changes in the thermal dynamics of Lake Sunapee, as documented by our projections, could lead to fundamental changes in other physical, chemical and biological processes in the lake. For example, stronger lake stratification can lead to increased hypolimnetic anoxia as a result of reduced mixing (*Jankowski et al., 2006*; *Piccolroaz, Toffolon & Majone, 2015*), leading to increases in greenhouse gas emissions and fluxes of nutrients and carbon from the sediments into the water column (*Hounshell et al., 2021*; *Bartosiewicz et al., 2021*; *Carey et al., 2022a*). Thermal habitat availability for a range of organisms (*e.g.*, coldwater fishes) is likely to decrease under a warmer thermal regime (*Hansen et al., 2017*; *Stetler et al., 2020*; *Kraemer et al., 2021*), especially in the northern hemisphere (*Comte & Olden, 2017*). Warmer temperatures and changes in ice cover may promote the dominance of cyanobacteria in the phytoplankton community (*Wagner & Adrian, 2009*; *Markensten, Moore & Persson, 2010*; *Elliott, 2012*; *Janse et al., 2015*), leading to deleterious consequences for drinking water and fisheries (*Paerl & Paul, 2012*; *Rigosi et al., 2014*).

## Study challenges and opportunities for future work

Our study provides several opportunities to build upon in future research. First, while this study focuses on model selection uncertainty through the use of ensemble modeling, numerous other sources of uncertainty may be important to lake thermal projections. In particular, uncertainty within a lake or climate model, potentially due to parameters, initial conditions, or specific processes, was not explored in this study (*Pike et al., 2013*; *Huang et al., 2013*; *Kim et al., 2014*; *Page et al., 2018*; *Thomas et al., 2020*; *Carey et al., 2022b*), and may be important to lake thermal projections. In addition, we acknowledge that our parameter calibration was likely influenced by the availability of water temperature both over space and time. First, our observations of water temperature below 10.0 m depth are limited to <5 profiles per year, meaning that the calibration of our models may be biased towards surface waters. Second, due to the challenges of collecting high-frequency data leading up to and immediately following ice cover in Lake Sunapee, it is possible that our parameters are biased towards recreating spring, summer, and fall dynamics, potentially leading to greater lake model uncertainty during the winter. While we present lake thermal metrics calculated over the entire year during the projection period (2006–2099), parameters which are biased towards over open-water conditions could have magnified the projected changes in our results, as the highest increase in surface water temperatures in some regions has been found to occur from May to August (*Czernecki & Ptak, 2018*). Our temporal data availability also precluded us from fully examining uncertainty dynamics from October to May, which may have yielded different results from the open-water period during when our observations were focused (*Thomas et al., 2020*).

Lastly, we emphasize that our study presents an "ensemble of opportunity", meaning that our results are inherently influenced by the specific models which were included in our analysis. As a result, our study findings are likely not exhaustive of the possible range of uncertainty due to variability among lake and climate models, and model performance is likely affected by the ways in which these specific models simulate thermal dynamics.

While we are confident that our results may extend to other lakes which are similar to Lake Sunapee, further testing of our ensemble methodology on additional lakes across geographic regions would be valuable to determining the scalability of our results. Lastly, while our high-frequency observations dating from 2007 to present allowed us to robustly calibrate and validate our lake models, we emphasize the value of continued long-term, high-frequency monitoring to quantity trends in Lake Sunapee's thermal metrics in response to changing climate (*e.g.*, *Desgué-Itier et al., 2023*). Altogether, important next steps building upon this study include examining uncertainty dynamics for additional water quality variables, seasons, and other sources of uncertainty not included in our study.

## CONCLUSIONS

Overall, our study demonstrates that uncertainty in lake thermal projections varies across depth, time, and RCP scenario for a north temperate, dimictic lake. We found that the dominant source of uncertainty varied among thermal metrics, with metrics that are more sensitive to atmospheric influence (*e.g.*, surface temperatures) dominated by differences among climate models, whereas within-lake metrics (*e.g.*, stratification duration) dominated by differences among lake models. However, the dominant source of uncertainty also varied over time within the projection period, with lake model selection uncertainty increasing more than climate model selection uncertainty. Additionally, uncertainty contributions appear to be different between short-term and long-term projections, calling for an improved understanding of uncertainty propagation over longer time scales and across metrics. Altogether, there was agreement in the overarching changes Lake Sunapee is likely to experience by the end of the century: Lake Sunapee's surface water will likely warm by 1–5 °C and will lead to 10–60 more days of summer stratification, as well as 20–75 fewer days of ice coverage annually, substantially changing the annual thermal structure of the lake. Overall, our findings regarding Lake Sunapee's potential future highlight the importance of robust calibration and validation, the use of an ensemble of lake and climate models, and full uncertainty partitioning for improved confidence in future climate projections of lake thermal dynamics.

## ACKNOWLEDGEMENTS

We gratefully acknowledge the Lake Sunapee Protective Association staff and members, especially June Fichter, Geoffrey Lizotte, Elizabeth Harper, John Merriman, Robert Wood, and Teriko MacConnell for sensor installation, data access, and for their long-term contributions to monitoring on Lake Sunapee, laying the framework to make this study possible. We thank Bethel Steele and Nicole Ward for data collation and modeling support on Lake Sunapee. We thank the Carey Lab for their helpful input on analyses and results during our study, as well as the Advanced Research Computing at Virginia Tech for providing computational resources.

### Funding

Funding was provided by the National Science Foundation (DBI-1933016, DBI-1933102), the Virginia Tech College of Science Luther and Alice Hamlett Scholarship, as well as the Lake Sunapee Protective Association (LSPA) and the LSPA-VT Calhoun Fellowship program. The funders had no role in study design, data collection and analysis, decision to publish, or preparation of the manuscript.

### Grant Disclosures

The following grant information was disclosed by the authors:
National Science Foundation: DBI-1933016, DBI-1933102.
Virginia Tech College of Science Luther and Alice Hamlett Scholarship, as well as the Lake Sunapee Protective Association (LSPA).
LSPA-VT Calhoun Fellowship program.

### Competing Interests

The authors declare there are no competing interests.

### Author Contributions

- Jacob H. Wynne conceived and designed the experiments, performed the experiments, analyzed the data, prepared figures and/or tables, authored or reviewed drafts of the article, and approved the final draft.
- Whitney Woelmer conceived and designed the experiments, prepared figures and/or tables, authored or reviewed drafts of the article, and approved the final draft.
- Tadhg N. Moore conceived and designed the experiments, performed the experiments, prepared figures and/or tables, authored or reviewed drafts of the article, and approved the final draft.
- R. Quinn Thomas conceived and designed the experiments, authored or reviewed drafts of the article, and approved the final draft.
- Kathleen C. Weathers conceived and designed the experiments, authored or reviewed drafts of the article, long-term data provisioning, and approved the final draft.
- Cayelan C. Carey conceived and designed the experiments, authored or reviewed drafts of the article, and approved the final draft.

### Data Availability

All code is available on Zenodo: Jacob Wynne, Whitney Woelmer, Tadhg Moore, R. Quinn Thomas, Kathleen C. Weathers, & Cayelan C. Carey. (2023). jacob8776/sunapee_LER_projections: release following revisions at PeerJ April 2023 (v2.0). Zenodo. https://doi.org/10.5281/zenodo.7821910.

The ensemble projections produced in this study are archived and available for analysis at Zenodo: Jacob H. Wynne, Whitney M. Woelmer, Tadhg N. Moore, R. Quinn Thomas, Kathleen C. Weathers, & Cayelan C. Carey. (2022). Ensembled projection outputs of Lake

Sunapee using multiple General Circulation models and Lake models [Data set]. Zenodo. https://doi.org/10.5281/zenodo.7232735.

The data required to run the analysis were downloaded from remote repositories and linked in the Zenodo code repository.

## Supplemental Information

Supplemental information for this article can be found online at http://dx.doi.org/10.7717/peerj.15445#supplemental-information.

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
