# Peer review of "Uncertainty in projections of future lake thermal dynamics is differentially driven by lake and global climate models"

_PeerJ, doi:10.7717/peerj.15445_

## Round 0.1 · original submission · Minor Revisions

Both reviewers appreciate the work you’ve put into this paper, and have a number of suggestions that I think would improve the paper’s clarity

Reviewer 1 ·

Basic reporting

The study goal was clearly stated in the manuscript. The literature review was adequate. The descriptions of data used for the study was clear. The structure is fine, but I found there were some redundancy, and the discussion is too long without in-depth discussion about the findings (especially new findings). The conclusion section can be shortened with the major findings of this study, considering the main goal of this study.

Experimental design

The research questions were well defined, but there are many parts that need better justifications regarding the model calibration. Especially, the use of a random sampling method is questionable. In addition, the essential details of the model calibration practices were provided. I left my comments on these in Additional Comments.

Validity of the findings

I agree with the findings, and the findings are useful. The authors may want to better contextualize the findings with other relevant studies. Again, the conclusion section can focus on the major findings.

Additional comments

Line 38: Please briefly elaborate on the “change over the next century.”

Line 146: There are GCMs (and their variants) more than four; please justify the selection of the four GCMs. I am not sure if the number of GCMs selected or used in this study is large enough to represent all variations in future climate projections.

Line 164: I think this is an important part of the method. Please elaborate on the “ensemble of opportunity” here.

Lines 196 to 199: Please elaborate on the “modeling-grouping protocol.”

Line 269: Considering the hydrological variations of the lake, I am not sure the warm-up period of 180 days might be enough to “avoid errors associated with initial conditions.” Please justify the selection of the spin-up period of 180 days.

Lines 284 to 286: The authors used LHC sampling to explore the parameter spaces of the models, but I am not sure if LHC sampling could locate the global optimum or ones close to the global optimum with the limited number of iterations or parameter sets sampled in the calibration and if the parameter spaces of the models were explored by LHC sampling at the same (or similar) level. The number of parameter sets to be sampled should be dependent on the complexity of the parameter spaces (or the number of calibration parameters). Please describe the details of the LHC sampling method employed in the study, including the number of parameter sets sampled and the parameter ranges sampled.

Lines 292 to 297: I am not sure if the selection of the mean water column depth for the calibration is well justified here. The variables of interest for model calibration are usually selected considering the goal of the modeling.

Lines 308 to 312: I think the lake model selection uncertainty may include parameter uncertainty as parameter uncertainty is part of modeling uncertainty. However, I also understand that this study specifically focuses on the selection of lake models and climate models.

Lines 489 to 500: It is interesting to find that the contributions of the selection of climate and lake models changed over time. I think it will be very useful to discuss why this can happen.

Lines 507 to 510: I agree that it is interesting to find that they did increase quickly in the beginning of the projection period. Please discuss why this can happen.

Lines 572 to 581: The authors mentioned the temporal changes of the uncertainty contributions again but did not discuss the reasons why they changed over time. I think it may be important to investigate and discuss the reasons, which will help better understand the time-dynamic nature of the uncertainty. I think there might be a threshold air temperature that triggers or promotes thermodynamic processes, but I am not sure.

Lines 611 to 619: I think it may be useful to discuss why this study (or Thomas et al.’s study) demonstrates the “opposite effect.”

Lines 630 to 631: How can we reduce overall lake model uncertainty? It may be very useful if the authors can suggest ways to reduce the lake model uncertainty here.

Lines 659 to 662: It may be also important to consider parameter uncertainty when interpreting ensemble mean projections.

Lines 751 to 753: It may be very useful if the authors can make suggestions for improving confidence in future projections based on the study results.

Reviewer 2 ·

Basic reporting

Overall the paper is well-written with clear language, a clear writing style, and clear structure.

Experimental design

Overall I think the methods make sense. One area that I think could have room for improvement is in additional ways of quantifying uncertainty. As stated by the authors, the Feldbauer et al 2022 paper uses a different method to quantify uncertainty. Why did the authors calculate uncertainty the way they did and what impact could this have had on results? Were other methods for analyzing uncertainty chosen?

Regarding the lake model set up, it would be helpful to include a little more information (perhaps in a supplement) on model inputs and outputs including what the hypsography curve was for model input and information about inflows and outflows (if used) within the model. Lake Sunapee has a complex bathymetry to be using a one dimensional model and I question generally whether it is an appropriate choice for that particular Lake. That doesn't negate the overall results of this study, which I do still think generally hold regardless of the complex bathymetry. But it would be good to include more of that information so others can potentially replicate and properly interpret results. Manual sampling sites and buoy sites are very different, so understanding the full picture of model set up and calibration would be useful in critiquing the uncertainty from model selection especially.

Validity of the findings

Findings are well stated and linked to original research question. The general overall finding that uncertainty in surface water metrics are linked to climate model selection and mixing and bottom water metrics linked to model selection and model processes is not surprising, but it is good to have this documented clearly in the literature.

I think the paper would be improved by digging in a little more to the modeling details and understanding how model selection does or does not play a role in uncertainty rather than lumping all the models together. As stated previously, this lake is particularly tricky to model given the complex bathymetry and assumptions that may need to be made.

Authors do make a note about the switch in climate vs lake model selection uncertainty to total uncertainty for some metrics. I wonder if you would see the same result using uncalibrated models or if you were to include differing calibration parameters in the lake model ensemble. One question that has come up before is whether calibrating models to current conditions negates their validity for simulating future climate conditions which are outside the range of current conditions. I think the paper would benefit in discussing this a little more and detailing whether some of the calibration variables may have had an impact on results of this study.

Additional comments

* Authors should define surface and bottom water temperatures - are these true surface and bottom temperatures from the model outputs or was there any depth averaging?
* Where is the lake hypsography from and how did you transfer the complex Lake Sunapee bathymetry to a one-d hysography curve for the models?
* is there inflow and outflow in the Lake? If so, can you detail how this was considered in the models?
* did you assume that the buoy and manual thermal profiles were equal since they were taken in different locations? Or did you do any work to combine those datasets for calibration? If this is detailed in other work, explicitly stating this and including the reference would be useful.
* Ice off is pretty hard to model because it's hard to capture some of the time-dependent drivers in ice formation. How did you account for this in your model and what was the variability in ice-on dates across models?
* How does the RMSE of models compare to the change seen over the historical period? Similarly, what are the trends over the historic period and how does the change in trend impact results and the calculation of "baseline" conditions for comparing the models?
* How does error propagation from not having observed data during the mixing and ice covered periods impact to include in calibration influence overall results? Could part of the model uncertainty simply be from poor model inputs?
* The paper discussed a little "outlier" models, but could the authors add any thoughts on whether ensemble approaches should exclude models which are not good physical fits either to the lake(s) being studied or to the observations used to calibrate models.
* I think the authors could expand a little on future work possibilities - e.g. do the results hold across lake types? Does model selection in the lake model ensemble influence results? You discussed some related to the RCP selection, but not as much for the lake model types used and I think this would improve the discussion section of the paper.

---

## Round 0.2 · accepted · Accept

Thanks for addressing all of the reviewers' comments, and congratulations!